

# The co-expression networks of differentially expressed RBPs with TFs and LncRNAs related to clinical TNM stages of cancers

Shuaibin Lian[1,*], Liansheng Li[2,*], Yongjie Zhou[1], Zixiao Liu[1] and Lei Wang[2]

[1] College of Physics and Electronic Engineering, XinYang Normal University, Xinyang, HeNan, China
[2] College of Life Sciences, XinYang Normal University, Xinyang, HeNan, China
[*] These authors contributed equally to this work.

Corresponding authors
Shuaibin Lian,
shuai_lian@xynu.edu.cn,
shuai_lian@qq.com
Lei Wang, wangleibio@126.com

## ABSTRACT

**Background**. RNA-binding proteins (RBPs) play important roles in cellular homeostasis by regulating the expression of thousands of transcripts, which have been reported to be involved in human tumorigenesis. Despite previous reports of the dysregulation of RBPs in cancers, the degree of dysregulation of RBPs in cancers and the intrinsic relevance between dysregulated RBPs and clinical TNM information remains unknown. Furthermore, the co-expressed networks of dysregulated RBPs with transcriptional factors and lncRNAs also require further investigation.

**Results**. Here, we firstly analyzed the deviations of expression levels of 1,542 RBPs from 20 cancer types and found that (1) RBPs are dysregulated in almost all 20 cancer types, especially in BLCA, COAD, READ, STAD, LUAD, LUSC and GBM with proportion of deviation larger than 300% compared with non-RBPs in normal tissues. (2) Up- and down-regulated RBPs also show opposed patterns of differential expression in cancers and normal tissues. In addition, down-regulated RBPs show a greater degree of dysregulated expression than up-regulated RBPs do. Secondly, we analyzed the intrinsic relevance between dysregulated RBPs and clinical TNM information and found that (3) Clinical TNM information for two cancer types—CHOL and KICH—is shown to be closely related to patterns of differentially expressed RBPs (DE RBPs) by co-expression cluster analysis. Thirdly, we identified ten key RBPs (seven down-regulated and three up-regulated) in CHOL and seven key RBPs (five down-regulated and two up-regulated) in KICH by analyzing co-expression correlation networks. Fourthly, we constructed the co-expression networks of key RBPs between 1,570 TFs and 4,147 lncRNAs for CHOL and KICH, respectively.

**Conclusions**. These results may provide an insight into the understanding of the functions of RBPs in human carcinogenesis. Furthermore, key RBPs and the co-expressed networks offer useful information for potential prognostic biomarkers and therapeutic targets for patients with cancers at the $N$ and $M$ stages in two cancer types CHOL and KICH.

## INTRODUCTION

Recent research has highlighted the importance of changes in RNA metabolism in the mechanisms of carcinogenesis, including long non-coding RNAs (lncRNAs). RNA transcription, maturation, transportation, stabilization, degradation, and translation are molecular processes that regulate the cell cycle, as well as cell survival. The keys to regulating RNA metabolism are a group of proteins called RNA-binding proteins (RBPs), which participate in many steps at the post-transcriptional regulation level, and thereby determine the fate and function of each transcriptional transcript in the cell (*Fu & Ares, 2014*; *Stefanie, Markus & Thomas, 2014*; *Moore & Proudfoot, 2009*). Furthermore, dysregulated expression of some RBPs can lead to disease, including neurological disorders and cancers (*Wang et al., 2018*; *Kechavarzi & Janga, 2014*). For instance, *PAIP1* is proposed as a novel prognostic biomarker by affecting breast cancer cell growth (*Piao et al., 2018*). *TRNAU1AP* has been confirmed to play an important role in the regulation of cell proliferation and migration via the PI3K/Akt signaling pathway (*Hu et al., 2018*). *SRPR* is reported to regulate keratinocyte proliferation by affecting cell cycle progression and tend to show high expression in epidermal keratinocytes (*Kim et al., 2016*). *RBMS3* has been found to inhibit breast cancer cell proliferation and tumorigenesis by inactivating the Wnt/β-catenin signaling pathway (*Yang, Quan & Ling, 2018*). In addition, overexpression of *RPL34* is suggested to promote malignant proliferation of non-small cell lung cancer (NSCLC) (*Yang et al., 2016*). Silencing *RPL34* plays a blocking role in cell proliferation and metastasis, but promoting cell apoptosis of oral squamous cell carcinomas (OSCCs) (*Dai & Wei, 2017*; *Liu et al., 2015*). Moreover, the splicing regulator *PTBP2* is suggested to control a network of genes involved in germ cell adhesion, migration, and polarity and is also very essential for neuronal maturation (Qin et al., 2014; *Molly, Leah & Donny, 2017*; *Leah, Sarah & Thomas, 2015*). However, some RBPs can act as tumor suppressors. For example, *ZNFX1-AS1* is reported to suppress HCC progression via regulating the methylation of *miR-9* (*Wang et al., 2016*). In addition, the silencing of *SRSF7* affects the expression of osteopontin splice variants and decreases the proliferation rate of renal cancer cells (*Boguslawska et al., 2016*). Finally, *RPL22/eL22*, as a cancer-mutated RBP, tend to be anti-cancer via regulation of the MDM2-p53 feedback loop (*Cao et al., 2017a*; *Cao et al., 2017b*).

Consequently, improving our understanding of the characteristics of RBPs and non-RNA-binding proteins (non-RBPs) is an essential step for understanding their roles in tumorigenesis. Even though recent studies have shown that RBPs are predominantly dysregulated in cancers relative to normal tissues (*Wang et al., 2018*; *Bobak & Sarath, 2014*). But, the intensity of dysregulation of RBPs in different cancers is still need to be investigated. Furthermore, cancer is a complex genetic disease. The different developmental stages of cancer indicate different degrees of severity. The early detection of cancer is linked to improved survivorship. The tumor-node-metastases (TNM) system, formed by the Union for International Cancer Control (UICC) and the American Joint Committee on Cancer (AJCC), is the most widely used cancer staging system (*Sobin, Gospodarowicz & Wittekind, 1992*; *American Joint Committee on Cancer, 2010*). The TNM system provides information

about the prognosis of the disease of patient based on pathologist evaluations of resected specimens. This information is also generally used to plan cancer treatment regimens. Due to the dysregulation of RBPs in many cancers, it is also interesting to determine whether there is an intrinsic relevance between RBP dysregulation and the developmental stages of cancers.

Transcription factors (TFs) perform the first step in interpreting the genome by recognizing specific DNA sequences to control transcription and gene expression (*Lambert et al., 2018*). As the unique gene class, TFs represent the proteins whose binding sites are affected by various regulatory variants in DNA. An accumulating genome-wide association study (GWAS) shows that the mutations of TFs or TF-binding sites are closely related to many human cancers (reviewed in *Deplancke, Alpern & Gardeux, 2016*), such as gastric cancer (*Yin et al., 2017*), liver cancer (*Cao et al., 2017a*; *Cao et al., 2017b*), prostate cancer, colorectal cancer (*Saijo et al., 2016*) and breast (*Humphries et al., 2017*) cancer. Several studies have emerged to identified the regulatory mechanisms and interactions (*Drissi et al., 2015*; *Zhang, Shen & Cui, 2019*). Yet, in most cases, we still do not know how to interpret the regulatory interactions between RBPs and TFs. Furthermore, long non-coding RNAs (lncRNAs)-transcripts of greater than 200 nucleotides are of vital importance in transcriptional and post-transcriptional levels (*Ishizuka et al., 2014*). Several evidences have demonstrated that the expression of lncRNAs is closely related to human diseases (*Yuan et al., 2017*; *Vergara et al., 2012*), such as viral infections, neurological disease and cancers (*Gibb, Brown & Lam, 2011*). In particular, the expression levels of lncRNAs in tumor tissues are significantly different compared with the normal tissues (*Gloss & Dinger, 2015*). Moreover, lncRNAs have merged as the novel biomarkers in several diseases diagnosis and targets for therapeutics (*Zhou et al., 2013*). It is well established that transcription factors and long non-coding RNA have played a central role in the genetics of human diseases (*St. Laurent, Wahlestedt & Kapranov, 2015*). Yet, up to now, we are far from being able to know the regulatory interactions and mechanisms of RNA-binding proteins (RBPs) with TFs and lncRNAs. For example, whether there is an interaction between RBPs and TFs in the process of carcinomatosis and what types of TFs affect the regulatory interaction most? In which stage of cancers do RBPs affect the expression of lncRNA the most?

To comprehensively characterize intensity of dysregulated RBPs from many human cancers (*Wang et al., 2018*) and construct their interactive networks with TFs and lncRNAs related to TNM stages, we first assessed the deviations of gene expression levels of RBPs and non-RBPs in 20 types of cancerous and normal (control) tissues, respectively (*Chang et al., 2013*) and analyzed the biological and molecular functions of dysregulated RBPs. Second, we analyzed the relationship between RBP dysregulation and TNM system clinical data of 5,093 patients across 13 types of cancers. We found out two types of cancers—cholangiocarcinoma (CHOL) and kidney chromophobe (KICH)—show a significant relationship between RBP dysregulation and TNM stage information. Third, we constructed the interaction networks for dysregulated RBPs related to TNM stage information and TFs of CHOL and KICH, respectively. Fourth, we constructed the interaction networks for key RBPs related to metastases (M) stage and lncRNAs for CHOL and KICH, respectively (see Fig. 1, which outlines the computational workflow, and
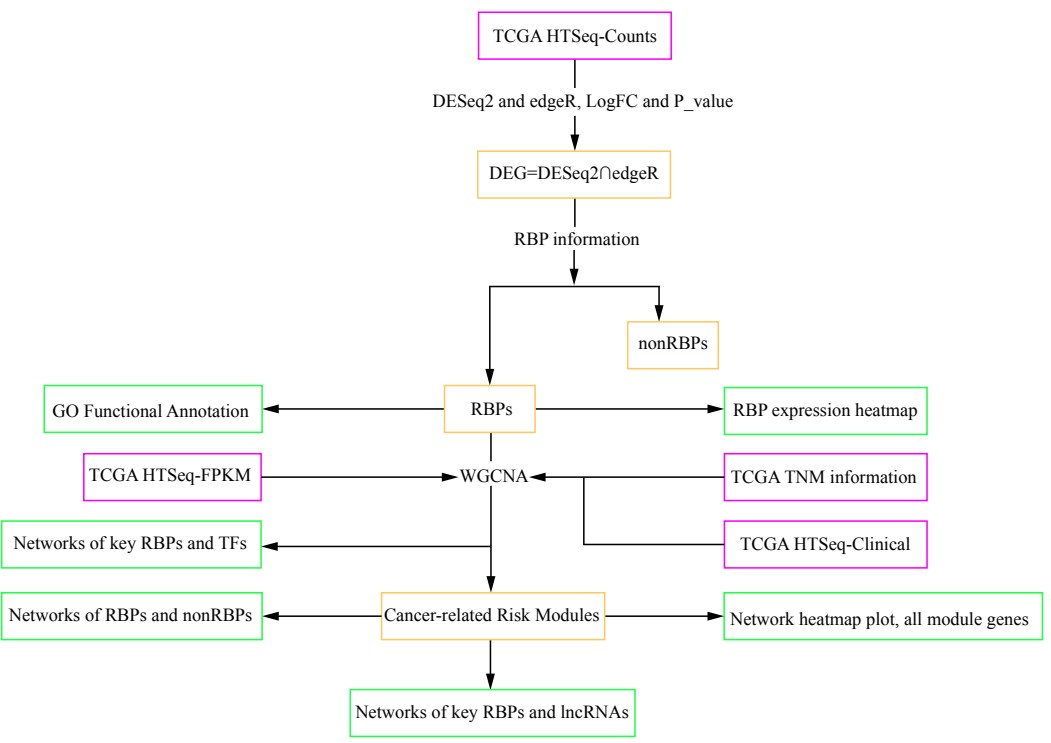

**Figure 1 Workflow chart showing the different steps presented in this study.** The flow chart shows the acquisition and preparation of data (Pink box), differentially expressed genes analysis and module detection (light yellow box), networks construction and function analysis (pale green box). RBP, RNA-binding protein; TCGA, The Cancer Genome Atlas; TF, transcription factor; lncRNA, long non-coding RNA.

Materials and Methods). This enabled us to identify the key regulatory RBPs for both CHOL and KICH cancers.

# MATERIALS & METHODS

## Differential expression analysis

We downloaded the original data including count matrix, expression FPKM values and clinical information of 20 cancer types and paired normal tissues from TCGA data base using SangerBox tool (SangerBox, http://sangerbox.com/). Then, we applied edgeR (*Robinson, Mccarthy & Smyth, 2010*) and DESeq2 (*Anders, 2009*) to select differentially expressed genes (DEGs) from count matrix for each cancer type with parameter padj < 0.05, and |log2FC| > 1. Thirdly, according 1,542 RBP genes information (*Stefanie, Markus & Thomas, 2014*), we divided the DEGs into two types for each cancer types: RBPs and non-RBPs. The expression FPKM values were used to compute the degree of dysregulation of DEGs and construct the co-expression networks. Clinical TNM information for 5,093 patients were used to co-expressed gene module detection. Our datasets contained 1,542 RBPs, 1,570 TFs and 4,147 lncRNAs; data acquisition information was presented in the Data Availability section. The 'pheatmap' package in R was used to generate a heatmap of differentially expressed RBPs shared by 16 cancer types in six systems based on log2-fold change values.

The Pearson coefficient (R) was used to judge the similarity of RBPs genes in different cancers occurring at related tissues have similar expression patterns. The gene cluster with $R > 0.5$ was considered as having the similar expression pattern. The gene ontology categorization analysis tool DAVID (*Huang, Sherman & Lempicki, 2009a*; *Huang, Sherman & Lempicki, 2009b*) was used to determine biological processes and molecular functions of DE RBPs.

## Standard deviation and proportion of dysregulation

In order to investigate the degree of dysregulation of RBPs, we computed the standard deviation of differentially expressed genes listed in Table 1 using the '*std*' function in 'MATLAB' version R2015b, respectively. In detail, we firstly computed the standard deviation of differentially expressed RBPs in 20 cancers and paired normal tissues (Fig. 2A). Similarly, we next computed the standard deviation of differentially expressed non-RBPs in 20 cancers and paired normal tissues (Fig. 2B).

The definition of the standard deviation is as follows:

$$\sigma = \frac{1}{N} \sqrt{\sum_{i=1}^{N}(x_i - \mu)^2}.$$

Here, $N$ is the number of genes, $x_i$ isthe expression FPKM value of the $i$th gene, $\mu$ is the mean FPKM value of all $N$ genes. Larger $\sigma$ values represent gene expression values that deviate from the mean value to a greater degree, which is indicative of greater gene dysregulation. The standard deviation values for RBP expression data is presented in Table S1.

In order to more clearly see the expression differences of RBPs and non-RBPs in cancers and normal tissues, we further computed their relative deviations. In detail, we first computed the relative deviation of RBPs in cancers and paired normal tissues, and then we computed the relative deviation of non-RBPs in cancers and normal tissues (Fig. 2C). The relative proportion of deviation is computed using the following function : $p = \frac{\sigma_c}{\sigma_n}$. Here, $\sigma_c$ and $\sigma_n$ are the standard deviations of the gene expressions in cancers and normal tissues respectively.

Similarly, in order to investigate the expression differences of up- and down-regulated RBPs, we also computed the standard deviation of them in cancers and normal tissues, respectively (Figs. 2D, 2E). Finally, the mean expression values of up- and down-regulated RBPs in 20 cancers and paired normal tissues were computed and presented in Figs. 2F & 2G.

## Weighted co-expression network analysis

Weighted gene co-expression network analysis (WGCNA) (*Peter & Steve, 2008*) is a comprehensive R package that summarizes and standardizes methods and functions for co-expression network analysis. Module detection function of WGCNA was used to detect the correlations between co-expression gene modules and the clinical TNM information for 13 types of cancers with default settings one by one. Then, the threshold of correlation coefficient $R > 0.5$ and statistical significance $P < 0.05$ was used to select

Lian et al. (2019), *PeerJ*, DOI 10.7717/peerj.7696

Peerj

**Table 1  Differentially expressed RBPs and non-RBPs across 20 types of cancers in seven systems.**

| System | Cancer | Cancer type | Total DEG genes | Total non-RBPs | Total RBPs | Up-regulated RBPs | Down-regulated RBPs |
|---|---|---|---|---|---|---|---|
| Landular system | PAAD | Pancreatic adenocarcinoma | 260 | 259 | 1 | 1 | 0 |
| | THCA | Thyroid carcinoma | 4,804 | 4,765 | 39 | 14 | 25 |
| | PRAD | Prostate adenocarcinoma | 4,777 | 4,722 | 55 | 34 | 21 |
| | HNSC | Head and neck squamous cell carcinoma | 7,421 | 7,335 | 86 | 32 | 54 |
| Respiratory system | LUSC | Lung squamous cell carcinoma | 12,235 | 11,972 | 263 | 54 | 209 |
| | LUAD | Lung adenocarcinoma | 9,004 | 8,859 | 145 | 35 | 110 |
| Alimentary system | READ | Rectum adenocarcinoma | 8,620 | 8,430 | 190 | 60 | 130 |
| | COAD | Colon adenocarcinoma | 8,954 | 8,773 | 181 | 44 | 137 |
| | STAD | Stomach adenocarcinoma | 9,037 | 8,932 | 105 | 39 | 66 |
| Urinary system | KICH | Kidney chromophobe | 9,926 | 9,758 | 168 | 76 | 92 |
| | KIRC | Kidney renal clear cell carcinoma | 11,478 | 11,370 | 108 | 37 | 71 |
| | KIRP | Kidney renal papillary cell carcinoma | 8,293 | 8,189 | 104 | 33 | 71 |
| | BLCA | Bladder urothelial carcinoma | 7,146 | 7,025 | 121 | 53 | 68 |
| Reproductive system | CESC | Cervical squamous cell carcinoma and endocervical adenocarcinoma | 4,302 | 4,154 | 148 | 57 | 91 |
| | UCEC | Uterine corpus endometrial carcinoma | 9,046 | 8,869 | 177 | 64 | 113 |
| | BRCA | Breast invasive carcinoma | 7,426 | 7,298 | 128 | 36 | 92 |
| Nervous system | GBM | Glioblastoma multiforme | 11,946 | 11,628 | 318 | 118 | 200 |
| | PCPG | Pheochromocytoma and paraganglioma | 4,717 | 4,560 | 157 | 101 | 56 |
| Liver and gall system | LIHC | Liver hepatocellular carcinoma | 6,677 | 6,576 | 101 | 25 | 76 |
| | CHOL | Cholangiocarcinoma | 10,219 | 10,030 | 189 | 80 | 109 |

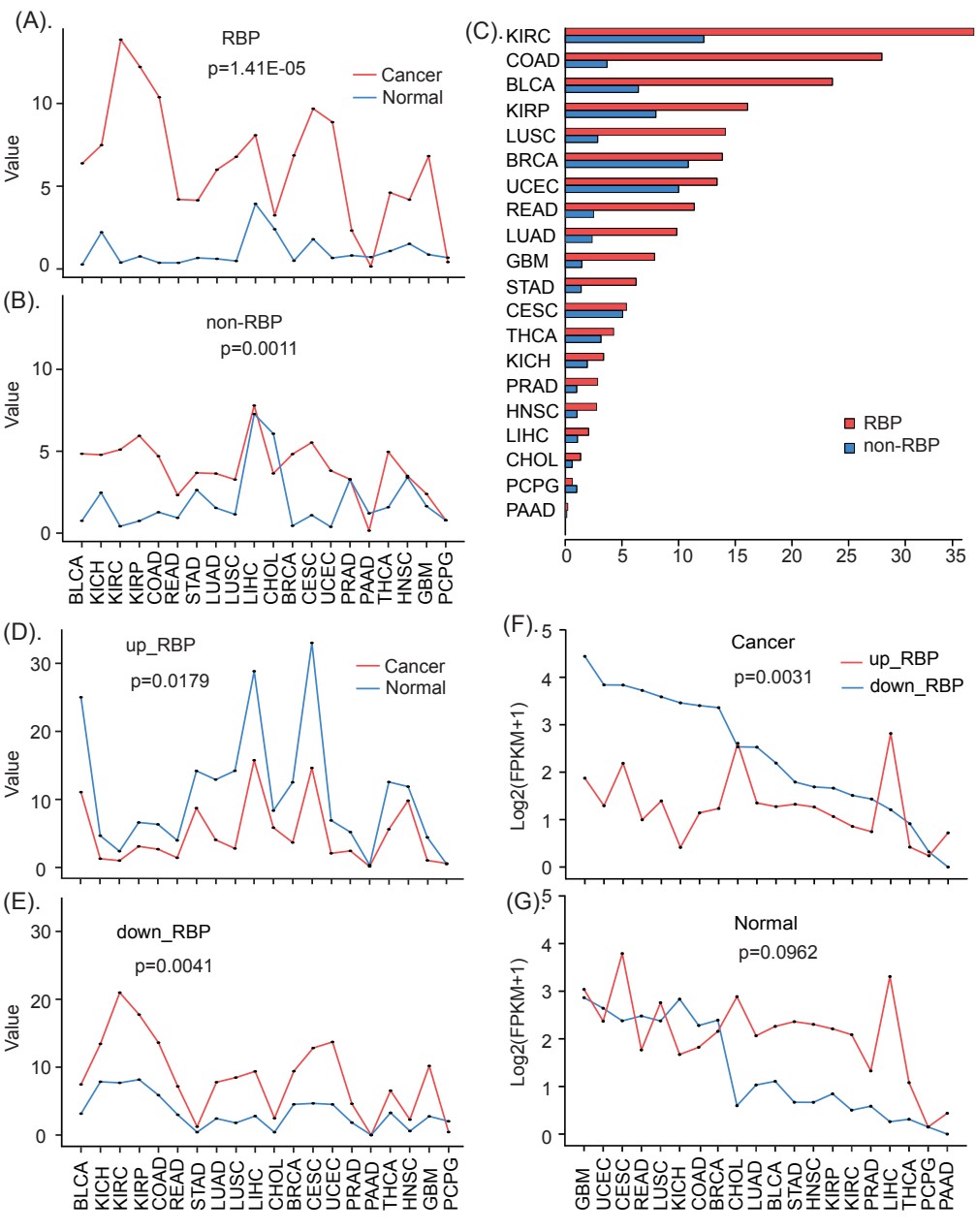

**Figure 2** **The different expression patterns of RBPs and non-RBPs in cancers and normal tissues.** (A) The standard expression deviation of RBPs in 20 types of cancers and normal tissues. (B) The standard expression deviation of non-RBPs in 20 types of cancers and normal tissues. (C) The relative expression deviation of RBPs and non-RBPs. (D) The standard expression deviation of up- regulated RBPs in cancers and normal tissues. (E) The standard expression deviation of down- regulated RBPs in cancers and normal tissues. (F) The expression FPKM values of up- regulated RBPs in cancers and normal tissues. (G) The expression FPKM values of down- regulated RBPs in cancers and normal tissues. Significance values calculated from the Mann–Whitney $U$ test are shown.

the cancer type whose TNM information related to co-expressed gene module. Genes with same expression pattern were clustered into one module and marked with same color. And then, the network construction module was used to construct co-expression networks for RBPs with TFs and lncRNAs in the CHOL and KICH cancer types, respectively. The detailed steps of constructing networks are as follows. Firstly, WGCAN network construction tool was used to generate the nodes and edges of genes by computing correlations of expression values. The nodes corresponded to genes, and the edges were determined by the pairwise correlations between the expression levels of genes. The corresponding called function in the R package was '*blockwiseModules*' and the parameters were set as follows: '*powers* = 10, *minModuleSize* = 30, *mergeCutHeight* = 0.25', other parameters we set to the default setting. Secondly, nodes with correlation $r < 0.5$ and edges with weighted threshold $<0.3$ were removed. Finally, the Cytoscape (https://cytoscape.org/) tool was used to plot the interactions using the nodes and edges of conserved genes.

## Clinical TNM information processing

Clinical TNM information for 5,093 patients exhibiting 13 types of cancers downloaded from the TCGA database presented in Table S4. Generally, in the TNM system, '*T*' refers to a primary tumor, '*T1~T4*' represents the severity of primary cancer according to the increase in tumor volume and the extent of involvement of adjacent tissues, and '*T0*' indicates no primary tumor. 'N' represents the tumor spreading to regional lymph nodes. '*N1~N3*' represents the degree of spreading according to the extent of lymph node involvement. '*M*' refers to tumor metastasis. No distant metastasis is expressed by '*M0*', and distant metastasis is expressed by '*M1*'. To investigate the correlation between TNM information and gene expression values using WGCNA, we converted the TNM information into a weighted matrix. For example, if TNM information for a patient was '*T1-N3-M1*', the corresponding weighted array is [1 3 1].

## Statistical methods

We used the Mann–Whitney $U$-test (function 'ranksum' in software 'MATLAB' version R2015b) to examine whether there is statistical significance between given two samples, the default significance level is 0.05 (*Lian et al., 2018*).

# RESULTS

## The deviations of expression levels of RBPs between cancerous and normal tissues

To investigate the degrees of dysregulation of RBPs in many human cancers, we computed the standard deviations of gene expression levels of RBPs and non-RBPs in 20 types of cancerous and normal (control) tissues, respectively. These results indicate that, relative to normal tissue, RBPs show greater variation in expression in almost all types of cancer ($P < 0.05$, Mann–Whitney $U$ test, Fig. 2A). In contrast, non-RBPs show considerably less variation in expression in cancers relative to normal tissues ($P < 0.05$, Mann–Whitney $U$ test, Fig. 2B). These results indicate that RBPs show a greater degree of dysregulated expression than non-RBPs in almost 20 types of cancers. Furthermore, the degrees of

dysregulation of RBPs in different cancer types are significantly different. In particular, RBPs show severely dysregulated expression in eight types of cancers, including BLCA, LUAD, STAD, READ, LUSC, GBM and COAD. For these cancers, the corresponding relative proportions of dysregulation relative to normal tissues are 365.5%, 415.8%, 446.4%, 456.3%, 494.5%, 540.9% and 759.1%, respectively (Fig. 2C, Table S1). Interestingly, up-regulated RBPs show a smaller standard deviation in expression in cancer tissue than in normal tissue ($P < 0.05$, Mann–Whitney $U$ test, Fig. 2D); However, down-regulated RBPs show considerable differences in the standard deviation of their expression in cancer tissue relative to normal tissues, especially in KICH, KIRC, KIRP, COAD, LUSC, CESC, UCEC, and GBM ($P < 0.05$, Mann–Whitney $U$ test, Fig. 2E). This may indicate that down-regulated RBPs are more dysregulated in cancers than are up-regulated RPBs. Finally, biological process enrichment analyzing indicates both up- and down-regulated RBPs were highly enriched in the processes, such as rRNA metabolism, nuclear-transcribed mRNA catabolism, and ncRNA processing (Fig. S1). In addition, up-regulated RBPs were also enriched in mitochondrial gene expression and in the regulation of mRNA metabolic processes, while down-regulated RBPs were enriched in ribosome biogenesis and rRNA processing.

## Up- and down-regulated RBPs show opposite expression patterns in cancer and normal tissue

To investigate cancer-specific differences in RBP expression, we analyzed the standard deviations and mean expression values of up-regulated and down-regulated RBPs in 20 types of cancers and in normal tissues (see methods and materials). Our results for all 20 cancer types suggests that, compared to normal tissues, up- and down-regulated RBPs show opposite patterns of expression in almost all cancers; what's more, down-regulated RBPs tend to show the larger expression deviations in cancers than up-regulated RBPs ($P < 0.05$, Mann–Whitney $U$ test, Figs. 2D, 2D). In particular, in almost all types of cancers, down-regulated RBPs show larger expression values than up-regulated RBPs. Furthermore, the expression deviations of up-regulated RBPs in cancers are lower than in normal tissues. However, the expression deviations of down-regulated RBPs are considerably greater in cancer tissue than in normal tissue; this is especially true for BLCA, GBM, HNSC, LUAD, LUSC, STAD, BRCA, KICH, READ, and UCSC ($P < 0.05$, Mann–Whitney $U$ test, Figs. 2E, 2G). These results suggest that down-regulated RBPs show a severer dysregulation in cancers than up-regulated RBPs. Furthermore, the expression pattern of up- and down-regulated RBPs in 20 types of cancers are the opposite of those found in normal tissues; what's more, molecular functions enrichment analyzing indicates that both up- and down-regulated RBPs showed enrichment in: catalytic activity acting on RNA, mRNA and mRNA 3′-UTR binding, nuclease and ribonuclease activity, and single-stranded RNA binding. In addition, we found enrichment in translation factor activity and RNA binding for up-regulated RBPs, and in catalytic activity acting on tRNAs for down-regulated RBPs (Fig. S1). In addition, we also were able to identify which specific biological processes and functions were regulated by up- and down-regulated RBPs, which may reveal how the normal processes of cells can be altered in a way that leads to cell carcinomatosis. These
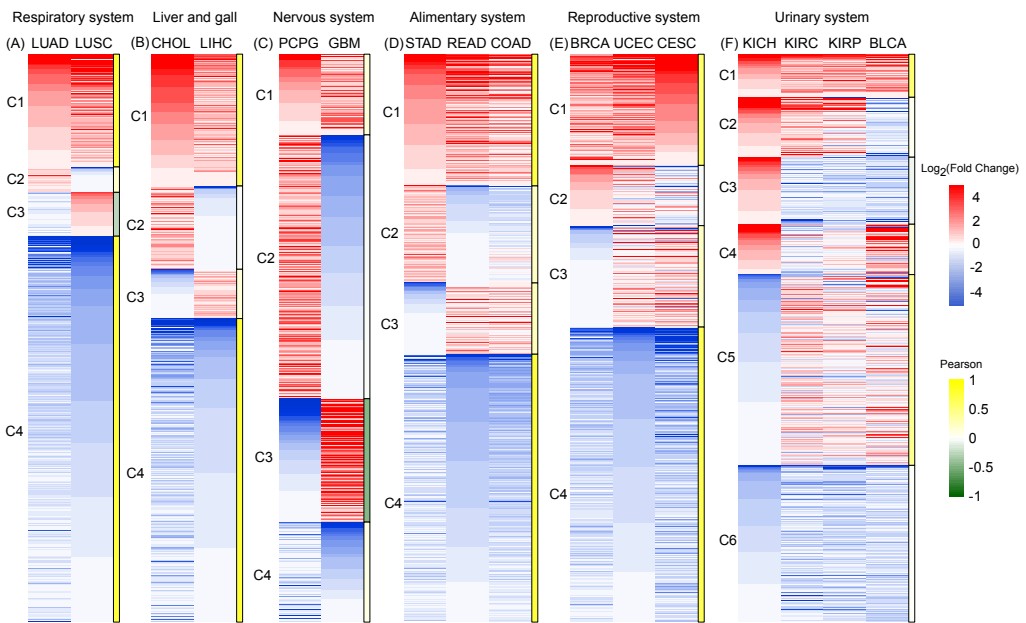

**Figure 3** **An expression heat map of 801 differentially expressed RBPs shared by 16 cancer types in six systems.** (A) Respiratory system. (B) Liver and gall. (C)Nervous system. (D) Alimentary system. (E) Reproductive system. (F) Urinary system. Red and blue represent high and low expression. The right column of each system is the Pearson coefficient $R$ of the corresponding cluster. Gene cluster with $R > 0.5$ was marked with dark yellow, which represents the similar expression pattern.

results provide a new insight into understanding the roles of up- and down-regulated RBPs in the process of cell carcinogenesis.

## RBPs of different cancers in same system have a similar expression profile

To gain clearer insight into RBP expression in cancerous tissues, we analyzed expression heatmaps of 16 types of cancers, except four types of glandular cancer (PAAD, THCA, PRAD, and HNSC). Because the number of DE RBPs in these four types of cancers are too small (Table 1). We divided DE RBPs into six scale systems according to type (i.e., organization of canceration) and analyzed the expression profiles of 801 DE RPBs shared by all 16 cancers by system. In each system, we divided RBPs into different types according to their expression values and Pearson coefficient $R$. Gene clusters with $R > 0.5$ were considered as having the similar expression pattern, which was shown in heatmaps. This was true for all cancers except two cancers of the nervous system. The corresponding gene lists of RBPs relevant for each system are presented in Table S2.

In the respiratory and liver and gall systems (Figs. 3A, 3B), two pairs of cancers in same systems, LUAD and LUSC, CHOL and LIHC, show very similar RBP expression profiles. Among these two cancer systems, C1 and C4 genes showed the co-expression patterns of RBP for high expression and low expression respectively ($R > 0.5$). The proportions of co-expressed RBPs in the respiratory and liver and gall systems were 88.8% and 76.8%,

respectively. However, RBP expression in two cancers of the nervous system (PCPG and GBM) showed opposite expression patterns (Fig. 3C). The proportions of highly expressed RBPs in PCPG and GBM cancer tissue were 60.5% and 36.1%, respectively. Highly expressed RBPs in PCPG showed a lower degree of expression in GBM, while highly expressed RBPs in GBM showed lower expression in PCPG. The proportion of RBPs showing opposing patterns of expression in PCPG and GBM was 68.2% ($R < 0.5$). In systems containing three types of cancers (i.e., the alimentary and reproductive systems) (Figs. 3D, 3E), the proportion of co-expressed RBPs decreased slightly, reaching 70.4% and 71.5% ($R > 0.5$), respectively. In the urinary system, which was affected by four types of cancers, the proportion of co-expressed RBPs reached its minimum value of 35.3% (Fig. 3F). Of the cancers of the urinary system, two (KIRC and KIRP) showed the strongest degree of RBP co-expression reaching 85.6% ($R > 0.5$). Taken together, our results suggest that RBP expression in different cancers in similar tissues have similar expression profiles.

## Co-expressed gene regulatory networks correlate with clinical TNM stage

Next, we investigated whether expression patterns are closely related to developmental stages of different cancers and constructed the co-expression networks for DE RBPs and non-RBPs using module detection and network construction tools of WGCNA (*Peter & Steve, 2008*). Results showed that for two types of cancers—CHOL and KICH—clinical TNM stage information was closely related to patterns of gene expression (Fig. 4). The results for the other 11 cancer types did not satisfy the threshold (Methods and Material, Figs. S2–S7).

In terms of module detection, we identified three modules (orange, green and red) related to the cancer metastasis stage (*M*-stage), as well as two modules (royal blue and red modules) closely related to the regional lymph node stage (*N*-stage) for CHOL (Fig. 4A). The red module was consistently in both the M and N stages. The gene list for each corresponding module is presented in Table S3. Furthermore, we found 10 differentially expressed RBPs in these three modules. These included six RBPs (*ACO1, PPARGC1A, PUS7, KHDC1, ELAVL3,* and *BICC1*) in the green module, two RBPs (*PANBP17* and *HNRNPA1*) in the royal blue module, and two RBPs (*DCPS* and *C2orf15*) in the red module. For KICH, we found that the royal blue module was most closely related to the *M*-stage, with a correlation coefficient of 0.96 ($P < 4e-15$), while the red and green modules were closely related to the *N*-stage (Fig. 4B). The gene list for each corresponding module is presented in Table S3. Furthermore, we found seven DE RBPs related to different developmental stages of cancer. These included four RBPs (*TNRC6A, MECP2, ZCCHC14,* and *POLR2F*) in the green module and three RBPs (*TDRD1, TDRD9,* and *CELF4*) in the red module. The regulatory networks of CHOL (Fig. 4C) revealed that (1) in each sub-network, one RBP interacts with almost all non-RBPs, suggesting that these RBPs are key regulators for each module; (2) the RBPs in different sub-networks interact with each other, indicating that they work together to regulate the corresponding developmental stage of the cancer. (3) we found that most key RBPs are down-regulated, the proportion is 80%. In addition, in the red module, two RBPs—including one down-regulated RBP (*C2orf15*)

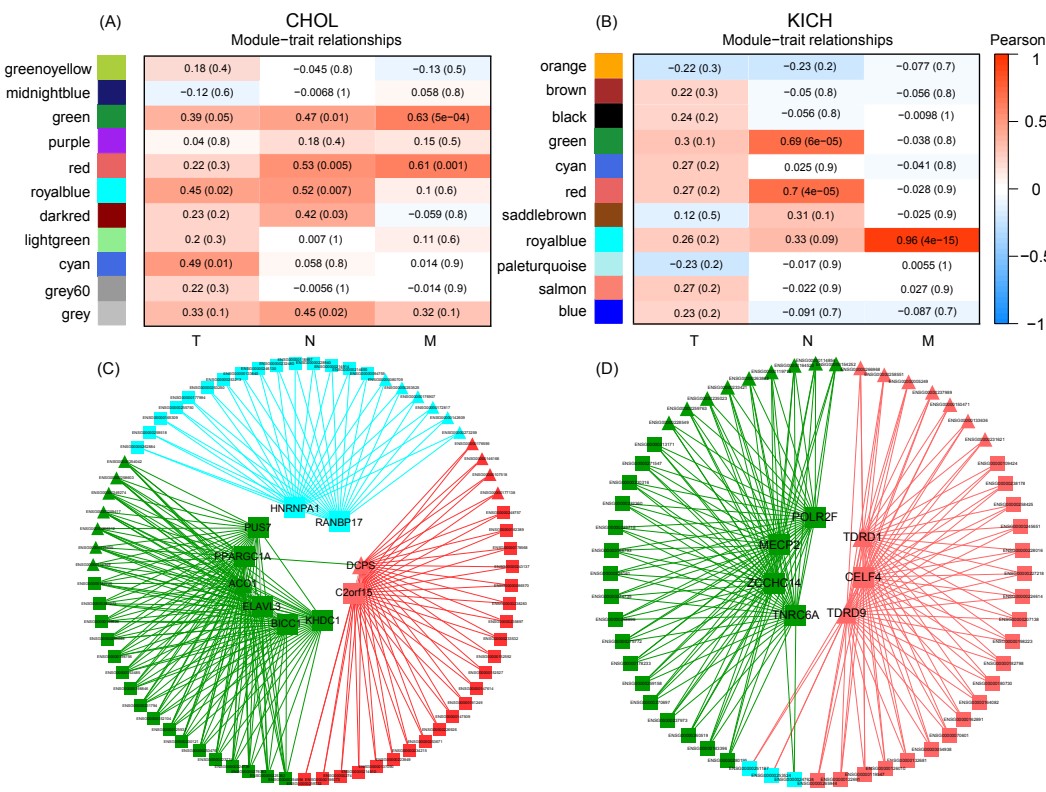

**Figure 4 The co-expression modules detection of CHOL and KICH, respectively.** (A) and (B) Co-expression modules of CHOL and KICH correlated with their clinical TNM stage information, respectively. Different color represents the gene modules with different expression pattern. The first, middle, and last column are detected gene modules related to *M*, *N*, and *T* stages, respectively. The two numbers in each module "*a(b)*" represent coefficients of co-expression and statistical significance, *a* is the co-expression coefficient and *b* is the corresponding *P*-value. The modules with coefficient larger than 0.5 and *P*-value smaller than 0.05 were considered as the related modules. (C) and (D) Co-expression networks of RBPs and non-RBPs in gene modules (green, red and royal blue) related with *M*- and *N*-stages for CHOL and KICH, respectively. Inner circle are key RBPs, outer circle are the non-RBPs. Triangular represents the up-regulated genes, block represents the down-regulated genes.

and one up-regulated RBP (*CDPS*) also play such a regulatory role. For the regulatory networks of KICH, we also identified three sub-networks that corresponded to the *M*- and *N*-stages (Fig. 4D). First, we found that RBPs interacted with almost all non-RBPs in each module, indicating that RBPs are key regulatory factors of the genes in these modules. Second, we identified seven key RBPs in the three modules, of which five were down-regulated and two were up-regulated. In the green module, we identified four key RBPs—*TNRC6A*, *MECP2*, *ZCCHC14*, and *POLR2F*—all of which were down-regulated. In the red module, we identified two key up-regulated RBPs (*TDRD1* and *TDRD9*) and one key down-regulated RBP, *CELF4*. These results suggest that dysregulated RBPs play a key role in the regulation of the development of the CHOL and KICH *M*-stage, which may provide a new perspective for potential prognostic biomarkers and therapeutic targets for patients with cancers at *M* stages in two cancer types CHOL and KICH.

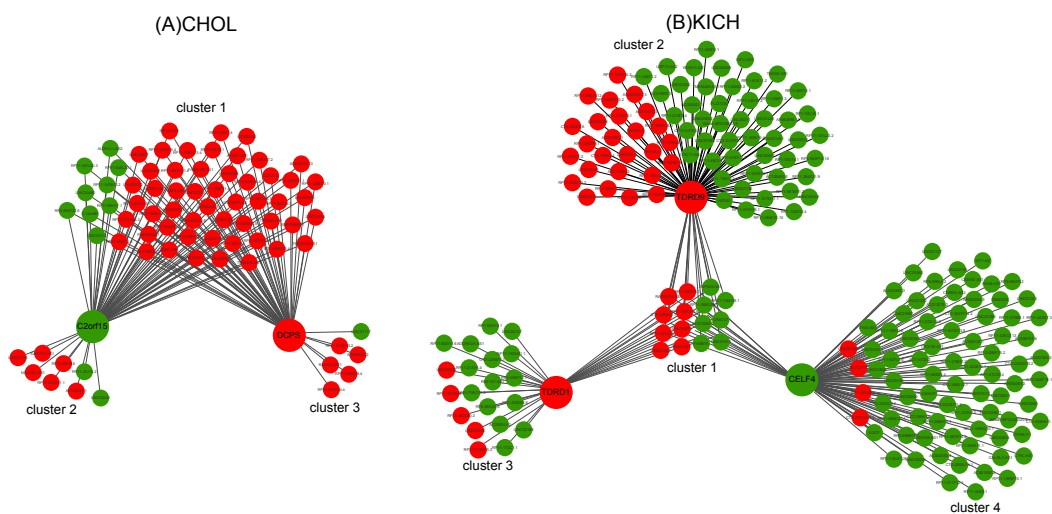

**Figure 5  The co-expression networks of key RBPs and lncRNAs.** (A) The co-expression network of two key RBPs related to *M* stage (*C2orf15* and *DCPS*) and lncRNAs for CHOL. (B) The co-expression network of three key RBPs (*TDRD1, TDRD9* and *CELF4*) related to *M* stage and lncRNAs for KICH. Red and green represent up- and down-regulated DE genes.

## The networks of key RBPs and lncRNAs for CHOL and KICH

To infer the potential regulatory mechanisms of lncRNAs with key RBPs related to *M* stages, we constructed the co-expression networks of key RBPs and differentially expressed (DE) lncRNAs for CHOL and KICH respectively and performed Gene Ontology and functional enrichment analyses.

Ten key RBPs and 2,943 DE lncRNAs were used to construct the co-expression network for CHOL. The resulting co-expression network consisted of two key RBPs (down-regulation *C2orf15* and up-regulation *DCPS*) and 75 lncRNAs, which were grouped into three clusters (Fig. 5A). There were 63 up-regulated lncRNAs, and the proportion of up-regulated lncRNAs in three clusters was 86%, 75%, 83.3%, which probably suggest that up-regulated lncRNAs have a greater interaction with RBPs in the process of metastasis of CHOL cells. Functional enrichment analyzing demonstrated that cluster 1 and cluster 3 had similar functions and mainly enriched in functional categories involved in gene silencing and negative regulation of translation, such as post transcriptional gene silencing, negative regulation of translation and cellular amide metabolic process, cellular response to dsRNA, miRNA metabolic process. Cluster 2 had some special functions and mainly enriched in such as positive regulation of mRNA catabolic process, cellular response to interleukin-1 and calcium ion, RNA destabilization.

Seven key RBPs and 1,204 DE lncRNAs were used to construct the co-expression network for KICH. The resulting co-expression network consisting of 3 key RBPs (up-regulated *TDRD1* and *TDRD9,* down-regulated *CELF4*) and 227 lncRNAs. There are 177 down-regulated lncRNAs in the network and were grouped into four clusters (Fig. 5B). The corresponding proportion of down-regulated lncRNAs was 46.7%, 65.3%, 75% and 95%, which probably suggest that down-regulated lncRNAs play more important roles in

interacting with key RBPs *TDRD1*, *TDRD9* and *CELF4* in the process of metastasis of KICH cells. Functional enrichment analysis demonstrated that cluster 1 shared by three key RBPs mainly enriched in dsRNA fragmentation and production of miRNAs involved in gene silencing by miRNA. Cluster 2 regulated by *TDRD9* mainly enriched in endoribonuclease and exon-exon junction complex, cluster 3 regulated by *TDRD1* mainly enriched in mRNA catabolic process and regulation of mRNA metabolic process, cluster 4 regulated by *CELF4* mainly enriched in transporting of RNA, mRNA and nucleic acids. Besides, cluster 1 and cluster 2 have some similar functions, such as telomere maintenance, histone mRNA and miRNA metabolic process, dosage compensation. Cluster 3 and cluster 4 have some similar functions, such as regulation of RNA stability, RNA localization and regulation of mRNA catabolic process.

These results provide a new insight into the understanding of the interactions of key RBPs with lncRNAs in the metastasis stage (*M* stage) of cancer cells.

### The co-expression networks of DEG RBPs and TFs for CHOL and KICH

To investigate the interactions of RBPs and TFs, we constructed the co-expression networks of DEG RBPs and TFs for CHOL and KICH, respectively. The key regulatory RBPs were those (ten for CHOL and seven for KICH) detected in above section.

The co-expression network of CHOL revealed two important insights. First, we found the five largest transcription factor families, they are *C2H2-ZF*, *Homeodomain*, *Nuclear receptor*, *bHLH* and *bZIP*, and the corresponding proportion is 37%, 17%, 9%, 9% and 9% (Figs. 6A, 6B), which interacted with almost all differentially expressed RBPs. This result indicates that these transcription factors tend to show a co-expression pattern with DEG RBPs, which further suggest that they play a major regulatory role in RBP post-regulatory levels for CHOL. Second, we also identified several special TFs related to up- or down-regulated RBPs for CHOL. For example, *Grainyhead, MADF, HMG-Sox* and *SAND* are specific transcription factors associated with down-regulated RBPs and *GTF2I-like, Myb-SANT, MADS box* and *CENPB* are specific transcription factors associated with up-regulated RBPs for CHOL. The co-expression network for KICH also revealed the following insights. First, we found that the four largest transcription factor families, they are *bHLH*, *bZIP*, *C2H2-ZF* and *Nuclear receptor*, the proportion is 54%, 9%, 7% and 7% (Figs. 6C, 6D). Notably, transcription factor *bHLH* interact with all RBPs and it accounts for more than half of all interacted transcription factors, which indicate that *bHLH* transcription factor probably involved in regulation of all differentially expressed RBPs for KICH. Second, the proportion of up- and down-regulated RBPs co-expressed with TFs is 57% and 43% respectively. But, the up-regulated RBPs tend to show more interactions with TFs (Fig. 6C). These results provide insights into understanding the mechanism of interaction between transcription factors and RBPs.

## DISCUSSION

RNA-binding proteins have been shown to be the key units to regulating RNA metabolism (*Fu & Ares, 2014*; *Stefanie, Markus & Thomas, 2014*; *Moore & Proudfoot, 2009*) and

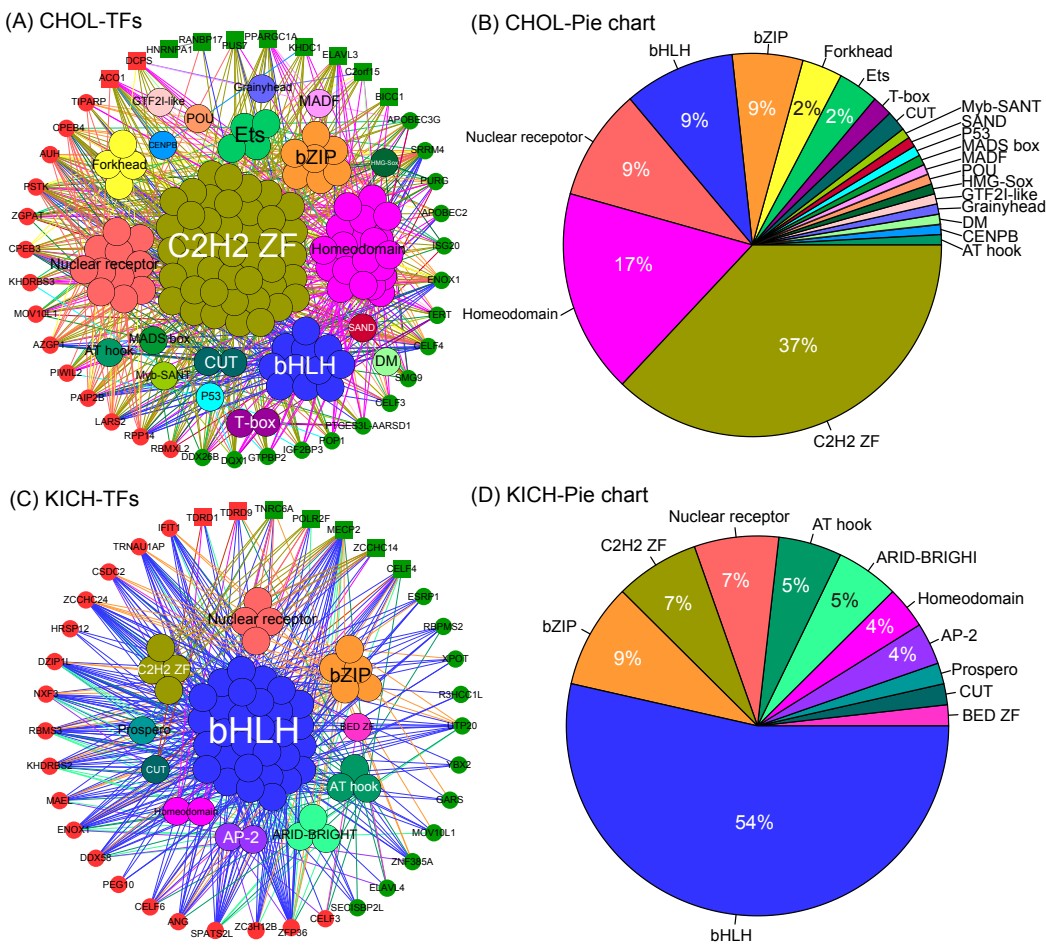

**Figure 6   The co-expression networks of DE RBPs and TFs.** (A) and (C) The co-expression networks of DE RBPs and TFs for CHOL and KICH respectively. (B) and (D) The corresponding pie chart of TFs co-expressed with DE RBPs for CHOL and KICH, respectively. Inner circle are key TFs, outer circle are DE RBPs. Red and green in outer circle represent up- and down-regulated RBPs. Block represents the key RBPs identified by module detection, circle dot represents other DE RBPs.

dynamically interact with both coding and noncoding RNA (*Kim et al., 2016*). Furthermore, recent studies have shown that RBPs are down-regulated in cancers (*Wang et al., 2018*), but the study of 16 tissues from 80 healthy individuals indicated that RBPs show the higher expression than non-RBPs (*Bobak & Sarath, 2014*). Consequently, we investigated the comprehensive expression differences of RBPs and non-RBPs simultaneously in cancers and normal tissues. Results indicate that RBPs are significantly dysregulated in cancers. In particular, recent studies have confirmed that RBPs show severely dysregulated expression in BLCA (*Kato et al., 2012*), LUAD (*Dong et al., 2018*), STAD (*Hapkova et al., 2013*), READ, LUSC (*Shi et al., 2017*), GBM (*Pavlyukov et al., 2018*) and COAD (*Saki et al., 2016*). Furthermore, up- and down-regulated RBPs tend to show opposite patterns of differential expression in cancers and normal tissues (Figs. 2D, 2E). Up-regulated RBPs show higher expression in normal tissues than down-regulated RBPs,

which is consistent with the results in *Bobak & Sarath (2014)*, but down-regulated RBPs show the significantly higher expression in cancers than up-regulated RBPs (Figs. 2F, 2G), which is consistent with results in (*Wang et al., 2018*). These results probably suggest a mechanism of RBPs in the process of carcinomatosis by which the up-regulated RBPs tend to show lower expression but down-regulated RBPs tend to show higher expression. Carcinogenesis is probably caused by the combined actions of the low expression of up-regulated RBPs and the high expression of down-regulated RBPs. This mechanism may be useful for understanding the roles of RPBs and the design of targeted drugs for cancer therapy.

We found 10 key regulated RBPs for CHOL (Seven down-regulated RBPs, *HNRNPA1, PANBP17, PUS7, KHDC1, ELAVL3, BICC1* and *C2orf15;* Three up-regulated RBPs, *ACO1 PPARGC1A and DCPS*) and seven key regulated RBPs for KICH (Six down-regulated RBPs, *TNRC6A, MECP2, ZCCHC14, CELF4* and *POLR2F;* Two up-regulated RBPs, *TDRD1* and *TDRD9*), respectively. Notably, recent studies have shown the importance of these key RBPs. For instance, multiple *PPARGC1A* transcripts are more abundant and CNS-specific in Parkinson's disease (PD) (*Soyal et al., 2019*). *KHDC1A* is highly expressed in oocytes and induces endoplasmic reticulum apoptosis (*Cai et al., 2012*). *Elavl3* is closely related to neurodegenerative diseases and play an important role in maintaining the axonal homeostasis of neurons (*Ogawa et al., 2018*). *HNRNPA1*, regulated by miR-503 and miR-424, is associated with breast cancer cell proliferation (*Otsuka, Yamamoto & Ochiya, 2018*). DCPS is very essential for acute myeloid leukemia cell survival by interacting with pre-mRNA (*Yamauchi et al., 2018*). *CELF4* plays an important role in brain development, the haploinsufficiency of *CELF4* is associated with autism disorders (*Barone et al., 2017*). *POLR2F* is significantly high expression in colorectal carcinomas (*Antonacopoulou et al., 2008*) and potential molecule in carcinogenesis. *TDRD1* is over-expressed in majority of 131 primary prostate tumors patients (*Xiao et al., 2016*). In all, these results have demonstrated that the key RBPs have played the important roles in other types of cell carcinomatosis and provide a new perspective for potential prognostic biomarkers and therapeutic targets for patients with cancers at the *N* and *M* stages in two cancer types CHOL and KICH.

## CONCLUSIONS

In this study, we analyzed detailed differences in the expression of RBPs and non-RBPs across 20 types of cancers and constructed the co-expression networks of dysregulated RBPs with TFs and lncRNAs for CHOL and KICH, respectively. Our results indicate that: (1) RBPs are dysregulated in almost all 20 cancer types compared with normal tissues, especially in BLCA, COAD, READ, STAD, LUAD, LUSC and GBM with proportion of deviation larger than 300% compared with non-RBPs in normal tissues. (2) Up- and down-regulated RBPs also show opposed patterns of differential expression in cancers and normal tissues. In addition, down-regulated RBPs show a greater degree of dysregulated expression than up-regulated RBPs do. (3) Clinical TNM information for two cancer types—CHOL and KICH—is shown to be closely related to patterns of differentially expressed RBPs (DE RBPs). (4) We constructed the co-expression networks of key RBPs between 1,570 TFs

and 4,147 lncRNAs for CHOL and KICH, respectively. By analyzing these networks, we identified ten key RBPs (of which seven were down-regulated and three up-regulated) in CHOL and seven RBPs (of which five were down-regulated and two up-regulated) in KICH. These key RBPs—and especially down-regulated RBPs—likely play important roles in cell carcinomatosis. This study lays the foundation for further efforts to understand the roles played by RBPs in human carcinogenesis and provides a new insight into identifying the potential prognostic biomarkers and therapeutic targets for patients.

## ACKNOWLEDGEMENTS

The authors thank three anonymous reviewers for their comments on the manuscript. The linguistic editing and proofreading provided by TopEdit LLC during the preparation of this manuscript are acknowledged.

### Funding

This work was supported by the National Natural Science Foundation of China (Grant. 61501392) and the Nanhu Scholars Program for Young Scholars of XYNU (Xin Yang Normal University). The funders had no role in study design, data collection and analysis, decision to publish, or preparation of the manuscript.

### Grant Disclosures

The following grant information was disclosed by the authors:
National Natural Science Foundation of China: 61501392.
Nanhu Scholars Program for Young Scholars of XYNU (Xin Yang Normal University).

### Competing Interests

The authors declare there are no competing interests.

### Author Contributions

- Shuaibin Lian conceived and designed the experiments, contributed reagents/materials/analysis tools, authored or reviewed drafts of the paper, approved the final draft.
- Liansheng Li performed the experiments, analyzed the data, contributed reagents/materials/analysis tools, prepared figures and/or tables, approved the final draft.
- Yongjie Zhou analyzed the data, contributed reagents/materials/analysis tools, prepared figures and/or tables, approved the final draft.
- Zixiao Liu analyzed the data, contributed reagents/materials/analysis tools, approved the final draft.
- Lei Wang conceived and designed the experiments, approved the final draft.

### Data Availability

Raw data is available in the Supplemental Files.

## Supplemental Information

Supplemental information for this article can be found online at http://dx.doi.org/10.7717/peerj.7696#supplemental-information.

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
