# Peer review of "The co-expression networks of differentially expressed RBPs with TFs and LncRNAs related to clinical TNM stages of cancers"

_PeerJ, doi:10.7717/peerj.7696_

## Round 0.1 · original submission · Major Revisions

Dear Dr. Lian,

It is my opinion, that your article requires a large number of Major Revisions.

I strongly suggest you to follow all requested changes they have asked for. Only in the event that you are able to address all changes I will consider your resubmission, although isn't guaranteed your article will be accepted.

Best regards
Giulia Piaggio

·

Basic reporting

The reviewer thanks the authors for their work and interesting principles and methods described in the manuscript. However, there are issues throughout the papers which must be addressed: Please find below my comments with the line in the text at which they are associated:

Generally, English needs to be improved throughout the text. There are few grammatic errors which must be changed. Please use an online grammar checker or simply your text editor.

Line 62: Please add a more recent review which describes the role of RBPs in disease.

Line 62: Please add a more recent review which describes the role of RBPs in disease.

Lines 63-75. The authors broadly described the role of RBPs pointedly adding references. However, the papers cited in any specific case only suggest a role, not demonstrate a role. The reviewer suggests the authors to shift the sentences to a more ‘suggested’, ‘potential’ role of the RBPs. Ex: Instead of using: X plays, X and Y block, Z inhibits, It would be better to switch in: X is suggested, Y has been proposed and so on.

Lines 82-88. I suggest the authors describe RBPs as dysregulated not specifying cases in which they have only been found downregulated. It is broadly accepted that RBPs are dysregulated in cancer, it is not necessary to specify only the cases of downregulation.

Lines 95-102. The authors need to reference more this passage and add some more specific published notions.

Lines 103-109: Add references and be more specific, citing well-described cases.

Lines 117-118: unclear

Lines 124: Please write the two types of cancer in long format and then acronyms

Line 409: The largest majority of RBPs studies show dysregulation not only downregulation. Please reformulate or add more details to this concept.

Line 437: The final statement must be reformulated. Demonstration implies experimental validations which have been not performed in this study.

Experimental design

The reviewer finds a lack or ambiguous description of the developed methods. The authors must clarify in the supplementary the source of the dataset. The reviewer finds not clear whether the authors downloaded only the raw gene expression data or the bam files and then run the RNA quantification. Moreover, in the case, they downloaded the raw counts they must address whether particular filtering of data took place to remove the noise of differential spectrum of expression, which may derive from characteristics of the samples and data processing to RNA quantification.

Line 137: How did the authors define the 2 groups of genes? How did they assign the genes to the 2 groups? This must be addressed in the methods.

Line 137: The authors did not clarify if they treated the differential genes in a cancer-specific way or a more general cancer vs normal (both pooled)

Line 171: The authors use a dubious terminology “The TNM system is simple to use and the prognostic accuracy between the different stages is quite good” which confer approximation the clinical annotation used. The reviewer suggests rephrasing the sentences in this paragraph adding more context and references of successful studies.

Line 177: What does WGCNA mean, please clarify?

Please add a figure which shows the computational workflow recapitulating the crucial points of the analysis.

Validity of the findings

Line 246: The final statement suggests that RBP genes in different cancers occurring at related tissues have similar expression patterns. Please add a figure showing that the gene clusters enrich for same pathways.
Line 263: Please add a reference to this statement.
Line 274: It is not clear how did the authors define and classify the role of genes in Mand N stages. Please clarify.
Line 299: please reformulate the statement adding some uncertainty or demonstrate what stated with experimental validations.
Fig 5. Figure not clear. The referee suggests summarising the data in bar plots per cluster.
Line 390, 399-400. Please reformulate or eliminate this very general and approximate sentence.
Line 401-404: The authors must provide more data showing which genes are potentially downregulated housekeeping or tissue-specific and which can be potentially linked to oncogenesis.

Reviewer 2 ·

Basic reporting

This manuscript by Shuai bin Lian and colleagues, titled “The co-expression networks of differentially expressed RBPs with TFs and LncRNAs related to clinical TNM stages of cancers”, presents a comprehensive analysis of the expression of RNA-binding proteins (RBPs), transcription factors (TFs) and lncRNAs in cancer. Authors take advantage of publicly available dataset from the TCGA studies to evaluate deregulated RBPs and subsequently analyze their association with information about the TNM classification of tumors. They identify in this way modules of RBPs associated with TNM specifically in two cancer types, CHOL (Cholangiocarcinoma) and KICH (Kidney renal clear cell carcinoma).
Next, they evaluate correlation of these RBPs with TFs and lncRNAs, identifying relevant networks related to clinical TNM classification in these cancer types.
Generally, the study is well conducted and described. Its relevance is mainly related to CHOL and KICH cancer types.
I include a few comments below for manuscript improvement.

Major points:
Page 9, lanes 132-135: In this description of the generation of list of differentially expressed genes it is not sufficiently clear whether you selected only genes differentially expressed (Tumor vs Normal) pervasively in all cancers and then you next analyzed their expression in the various types OR if you started from DEGs from each subtype. Please specify clearly.

In Figure 3A-B it not clearly described what raws represent. Please revise explaining more clearly what modules represent and how did you create these.

Line 277 page 14: …in each sub-network, one RBP interacts with almost all non-RBPs, suggesting that particular RBPs are key regulators for each model: please include description of what do you mean for “interact”. A clearer description of the relation between RBP and non-RBPs in each module needs to be also better described. Please just mention the method (WGCNA?) you used for correlation analysis and please mention if you also considered traditional correlation analysis for specific modules.
Generally, the point is that the text should be easily readable by scientists from all disciplines and I think you could try to better specify the analysis method applied at the beginning of each data/results section description (just mention the method), even if all analysis are well described in the Materials section.

Data presented in Figure 6 do not add much information in my opinion and this general analysis of RBP commonly deregulated in all cancer types could be moved to supplementary material and presented in the results text immediately after the first step of the analysis which led to the identification of these commonly DE RBPs.

Minor:
Speculations on the basis of the obtained results should be preferably included in the Discussion, while authors frequently include these in the results, Please revise.

Wang et al., 2018 is mentioned but not listed in the references

Experimental design

nc

Validity of the findings

nc

Additional comments

nc

Reviewer 3 ·

Basic reporting

In this study, Lian and co-authors explored the dysregulation of human RBPs across 20 cancer types compare to normal tissues. RBPs result differentially dysregulated between cancerous and normal tissues and RBPs belonging to the same system showed similar expression profile in different cancer types. The clinical TNM stage was related to pattern of RBPs expression in CHOL and KICH cancer types. Moreover, authors assess the co-expression networks of RBPs with Transcription factors and lncRNA in CHOL and KICH. Finally gene ontology analysis among 801 deregulated RBPs shared by 16 cancer types revealed that these genes are involved in some common biological processes and were enriched in some common biological functions.
RBPs are known to finely control gene expression and their dysregulation has been extensively reported to promote tumorigenesis. Thus, the evaluation of RBPs dysregulation and co-expression networks may help to elucidate the underlying biological mechanisms involved in tumorigenesis across cancers.
Despite the relevance of the topic, this manuscript is too descriptive and conclusions need to be expanded with additional information concerning the relevance of the obtained results. In Materials and methods section, the references of data analyses should be added. Moreover, some topics are unrelated, thus it is not easy to follow the rationale of the analyses. Finally, the authors' entire emphasis hinges on Figures 1, 3, 4 and 5 in the manuscript, which I cannot make any sense of. Text presents numerous contradictory statements and legends of these figures have to be more precise. All elements presented in the figures should be described in the legend or at least the text.

Experimental design

1. Figure 4. The link between Figure 3 and Figure 4 is not obvious. The link has to be clarified in the text.
2. Materials and methods section must be rewritten to be more detailed and accurate. References for data analyses are missing.

Validity of the findings

1. To corroborate the conclusions from Figure 1, the authors should assess whether the differences in each panel are significant. Thus, if applicable, statistical analysis should be added.
2. Lane 193-195: “In particular, RBPs show severely dysregulated expression in eight types of cancers, .... For these cancers, the corresponding relative proportions of dysregulation relative to normal tissues are....”. To reinforce the conclusion, the “relative proportions” of RBPs dysregulation should be shown for all cancer types (ie bar graph).
3. Figure 1B: To me, the rationale refers to this panel is not obvious. Up- and down-regulated RBPs are differentially expressed genes in tumors compare to normal tissues. While for normal tissues, what are they referring to? Moreover, how you can explain that the expression level of up-regulated RBPs in cancer is lower than in normal tissues? And vice versa. Please, indicate what the data refers to.
4. The main text concerning Figure 3 presents some contradictory statements that make it incomprehensible:
Lane 265: “…two RBPs (PANBP17 and HNRNPA1) in the green module”. In Table S3 these RBPs are in the royal blue module. Please verify.
Lane 265: “…and two RBPs (DCPS and C2orf15) in the royal blue module”. In Table S3 these RBPs are in the green module. Please verify.
Lane 270: “These included four RBPs (TNRC6A, MECP2, ZCCHC14, and POLR2F) in the green model…”. In Table S3 these RBPs are in the royal blue module. Please verify.
Lane 271: “….three RBPs (TDRD1, TDRD9, and CELF4) in the royal blue model”. In Table S3 these RBPs are in the green module. Please verify.
Lane 279: “Third, we found that among ten key (regulator) RBPs, six were down-regulated genes”. In Figure 3C, eight RBPs are resented as block (down-regulated genes). Please verify.
5. Figure 3, panels C and D. What are the different colors for? Please, indicate in the legend of panels C and D what the colors refer to. For example, in panel C, RBPs from Royal Blue module (RANBP17 and HNRNPA1) are indicated in light blue. RBPs from green (DCPS and C2orf15) and red (PUS7, PPARGC1a….) modules are indicated in red and green colors, respectively. This last representation is quite misleading. Please, if applicable, the green and red color in panel C should be inverted. Similarly, in panel D, RBPs from royal blue module are reported in green; while RBPs from green modules are indicated in red.
6. Figure 4 and 5: What are the different colors (green and red) for? Please, indicate in the legend also the meaning of circular, triangular and square dots.
7. To further strengthen the relevance of deregulated RBPs in CHOL and KICH pathogenesis, the prognostic impact (ie survival for patients) of co-expression networks should be addressed.
8. Please correct PANBP17 with RANBP17 in the main text
9. Lane 270 and 272: change “model” with “module”
10. Figure 1A, Panel a-d): To help reading comprehension, the title in the axis of abscissa should be added (ie cancer type).

---

## Round 0.2 · accepted · Accept

I am writing to inform you that your manuscript - The Co-expression Networks of differentially expressed RBPs with TFs and LncRNAs related to clinical TNM stages of Cancers - has been Accepted for publication. Congratulations!

# ·

Basic reporting

The reviewer would like to thank the authors for having submitted a new version of the manuscript totally revised. The reviewer appreciates the changes applied and do not rise any more concerns regarding language, references used, sentence formulation and final statements.

Experimental design

The reviewer appreciates the changes applied in the second version of the paper which strongly clarify the validity of the approach used. No more concerns are raised after reading the rebuttal and the second version of the paper regarding the computational workflow, which fulfils the technical standards in computational biology.

Validity of the findings

The authors clarified and changed what requested properly. The reviewer considers the conclusion and findings in line with the description and properly linked with the literature.

Additional comments

No further comments.

Reviewer 3 ·

Basic reporting

The authors improve the main text and Material and Methods sections. Figure legends are more precise.

Experimental design

All my comments concerning the experimental design have been answered.
Contradictory statements have been amended.

Validity of the findings

I believe that this work is now more solid.